# Accuracy and Clinical Relevance of Intra-Tumoral *Fusobacterium nucleatum* Detection in Formalin-Fixed Paraffin-Embedded (FFPE) Tissue by Droplet Digital PCR (ddPCR) in Colorectal Cancer

**DOI:** 10.3390/diagnostics12010114

**Published:** 2022-01-05

**Authors:** José Guilherme Datorre, Ana Carolina de Carvalho, Mariana Bisarro dos Reis, Monise dos Reis, Marcus Matsushita, Florinda Santos, Denise Peixoto Guimarães, Rui Manuel Reis

**Affiliations:** 1Molecular Oncology Research Center, Barretos Cancer Hospital, Barretos 14784400, Brazil; guilherme_datorre@hotmail.com (J.G.D.); anacarol.oak@gmail.com (A.C.d.C.); marianabisarro@yahoo.com.br (M.B.d.R.); guimaraes.dp@gmail.com (D.P.G.); 2Department of Pathology, Barretos Cancer Hospital, Barretos 14784400, Brazil; monise_reis@yahoo.com.br (M.d.R.); mmatsu@uol.com.br (M.M.); 3Department of Medical Oncology, Barretos Cancer Hospital, Barretos 14784400, Brazil; floalmeidasantos@yahoo.com.br; 4Department of Prevention, Barretos Cancer Hospital, Barretos 14784400, Brazil; 5Life and Health Sciences Research Institute (ICVS), School of Medicine, University of Minho, 4704553 Braga, Portugal; 6ICVS/3B’s—PT Government Associate Laboratory, 4704553 Braga, Portugal

**Keywords:** colorectal cancer, *Fusobacterium nucleatum*, ddPCR, qPCR, biomarker, prognostic, Brazil

## Abstract

The use of droplet digital PCR (ddPCR) to identify and quantify low-abundance targets is a significant advantage for accurately detecting potentially oncogenic bacteria. *Fusobacterium nucleatum* (*Fn*) is implicated in colorectal cancer (CRC) tumorigenesis and is becoming an important prognostic biomarker. We evaluated the detection accuracy and clinical relevance of Fn DNA by ddPCR in a molecularly characterized, formalin-fixed, paraffin-embedded (FFPE) CRC cohort previously analyzed by qPCR for *Fn* levels. Following a ddPCR assay optimization and an analytical evaluation, *Fn* DNA were measured in 139 CRC FFPE cases. The measures of accuracy for *Fn* status compared to the prior results generated by qPCR and the association with clinicopathological and molecular patients’ features were also evaluated. The ddPCR-based *Fn* assay was sensitive and specific to positive controls. *Fn* DNA were detected in 20.1% of cases and further classified as *Fn*-high and *Fn*-low/negative, according to the median amount of *Fn* DNA that were detected in all cases and associated with the patient’s worst prognosis. There was a low agreement between the *Fn* status determined by ddPCR and qPCR (Cohen’s Kappa = 0.210). Our findings show that ddPCR can detect and quantify *Fn* in FFPE tumor tissues and highlights its clinical relevance in *Fn* detection in a routine CRC setting.

## 1. Introduction

Colorectal cancer (CRC) is the second most commonly diagnosed cancer in women and the third most common in men, with 1.9 million cases and 935,000 deaths worldwide in 2020 [1]. According to the Brazilian National Cancer Institute, 20,520 cases in men and 20,470 cases in women are estimated per year from 2020 to 2022, being the second most common type of cancer in Brazil for both men and women [2]. Studies on CRC mortality showed increased rates worldwide, with Brazil being one of the countries where these rates were on the rise [3], with 9207 deaths from CRC in men and 9660 deaths in women, in 2017 [2]. There are several known CRC etiological factors, such as smoking, an unhealthy diet, high alcohol consumption, sedentary behavior, excess body weight, advanced age (>50 years), radiotherapy, a personal history of CRC or adenoma, inflammatory intestinal conditions, and genetic predisposition [4,5].

CRC is a heterogeneous disease characterized by various genomic and epigenomic alterations [6]. CRC diagnosis, therapy choice, and prognosis depend on the tumor classification and clinicopathological stage of the disease at the time of diagnosis [7,8]. For early-diagnosed CRC (stage 0, I, or II), the 5-year overall survival rate is >80%, but it decreases to <10% with a late diagnosis of metastatic cancer [9,10]. Identifying prognostic biomarkers to stratify patients according to their likelihood of clinical evolution and association with clinicopathological features of aggressive (or more indolent) behavior is crucial for accurate CRC management [11].

The accumulating data suggest that microbiota play a role in the etiology of CRC by influencing inflammation, DNA damage, and apoptosis [12]. With several enriched or diminished species, the gut microbiota are crucial in CRC tumorigenesis [13]. *Fusobacterium nucleatum* (*Fn*) is a major driver of CRC tumorigenesis [14,15,16,17]. *Fn* is a Gram-negative bacteria and a normal constituent of the human oral cavity, where its presence is associated with periodontitis [18]. *Fn* is implicated in the progression of advanced colorectal carcinoma and associated with clinical and molecular features, such as the proximal tumor location, *BRAF* mutation, MSI-high status, downregulating of antitumor T cell-mediated adaptive immunity, CRC staging, and a worse patient prognosis [14,19,20,21,22]. Therefore, the presence and levels of *Fn* DNA in tumor tissue can be used as a valuable CRC biomarker in a clinical setting [17,23,24,25].

The reported prevalence rates of *Fn* in CRC tissues vary widely from 9 to 87% [26]. These differences may be explained by differences in the type of tumor tissue analyzed (fresh-frozen, FFPE, and feces), methods of detection (FISH, qPCR, and NGS), sociodemographic features, such as the differences in the population and geographic location, and lifestyle factors, mainly dietary habits [26,27]. The most commonly used method to detect *Fn* is qPCR (quantitative real-time polymerase chain reaction) [17,22,26,27]. In a recent study using qPCR, our group detected *Fn* DNA in 23% of CRC fresh-frozen tissues, whereas, in the FFPE tumor tissue counterparts, the *Fn* was detected in only 5.8% of cases [19].

Importantly, FFPE is a routine material of laboratories worldwide, and thus the key tissue source for biomarkers analysis. Therefore, there is a need to develop a reproducible and sensitive method for *Fn* detection in FFPE. The droplet digital PCR (ddPCR) exhibits a much greater sensitivity than qPCR, and it allows the precise quantification of bacterial DNA copies without the requirement of calibration curves, simply by counting the number of successful amplification reactions and applying statistical adjustments to provide the number of bacterial DNA copies per reaction [28,29].

The present study evaluated the feasibility, accuracy, and clinical relevance of Fn DNA detection by ddPCR in a series of FFPE CRC samples. Moreover, the ddPCR levels of *Fn* were further compared with the *Fn* previously reported by quantitative PCR (qPCR). Our findings highlight the potential of ddPCR to identify intra-tumoral *Fn*-positive patients with a higher accuracy than qPCR, when used for FFPE tumor tissue.

## 2. Materials and Methods

### 2.1. Patient Population and Tissue Samples

This study analyzed a previously reported series of 139 patients with CRC treated at the Department of Colorectal Surgery of Barretos Cancer Hospital, Barretos, Brazil, between 2008 and 2015 [19,30]. FFPE slides were subjected to histological examination to confirm the diagnostic delimitation of tumor tissue by the Department of Pathology of Barretos Cancer Hospital and carefully macrodissected. Only tumor samples with the presence of at least 60% of tumor cells were included.

Patients’ clinical-pathological features, such as age, gender, location of the primary tumor, staging, and histological grade, as well as histo-molecular data on the expression of mismatch repair proteins (MLH1, MSH2, MSH6, and PMS2), the status of molecular microsatellite instability (MSI), and *BRAF* mutation and *Fn* quantification by qPCR, were previously reported [19,30]. Patients were followed for a median of 59.68 months (ranging from 2.37 to 104.97 months).

As previously reported, DNA from FFPE samples was isolated using the QIAamp DNA Micro Kit (Qiagen, Valencia, CA, USA) [19,30]. The Institutional Review Board approved this study at Barretos Cancer Hospital (Project number 1402/2017).

### 2.2. Detection of Fn DNA by Quantitative PCR (qPCR)

*Fn* detection from 139 FFPE tumor samples was previously analyzed by our group by quantitative PCR using TaqMan primer-probe sets (Applied Biosystems) for the 16S ribosomal RNA gene DNA sequence of *Fn* (nusG), and for the reference gene, *SLCO2A1*, as described [19]. Briefly, primer and probe sequences used were described by Mima et al. [21]. Each reaction contained 100 ng of genomic DNA run in duplicate in 20-µL reactions containing 1 × final concentration TaqMan Environmental Master Mix 2.0 (Applied Biosystems), 900 nM primers, and 500 nM probes for each target gene. Amplification and detection were performed with the QuantStudio 6 Flex Real-Time PCR System (Thermo Fisher Scientific) using 10 min at 95 °C and 45 cycles of 15 s at 95 °C and 1 min at 60 °C as reaction conditions. DNA from *F. nucleatum subsp. nucleatum Knorr* (ATCC 2558) was used as a positive control for all *nusG* runs. In FFPE, CRC cases with detectable *Fn* and the cycle threshold Ct) values in the quantitative PCR for *nusG* were normalized by *SLCO2A1* and used to calculate 2^−∆Ct^ values that were used to quantify the amount of *Fn* DNA in each sample as a relative unitless value (where ∆Ct = the average Ct value of *nusG*—the average Ct value of *SLCO2A1*) as previously described [31]. Samples were classified according to the amount of bacteria found as low/negative or high (*Fn*-low/negative, *Fn*-high), based on the median cut-point amount of *Fn* DNA in all samples with positive results (median = 6.0 × 10^−6^) [19].

### 2.3. Detection of Fn DNA by Droplet Digital PCR (ddPCR)

*Fn* detection by ddPCR was performed employing primer and probe sequences targeting a sequence of the *Fn* genome as previously reported [21,32]. The PCR amplicon length is 108 bp of NCBI reference sequence NC_003454.1 (*F. nucleatum subsp. nucleatum* ATCC 25,586 chromosome) [32]. The ddPCR reaction was performed with 20 ng of DNA, 1× ddPCR Supermix for Probes (No dUTP) (BioRad, Hercules, CA, USA), 0.25µM of each primer, and 0.125µM of the probe in a total volume of 20µL followed by droplet generation using an automated droplet generator (Bio-Rad Laboratories, Hercules, CA, USA). Cycling conditions included preheating at 95 °C for 10 min followed by 40 cycles of denaturation at 94 °C for 30 s, annealing at 60 °C for 60 s, and a final heating at 98 °C for 10 min [32]. After amplification, the PCR plate was transferred to a QX100 droplet reader (Bio-Rad Laboratories, Hercules, CA, USA), and fluorescence amplitude data were obtained by QuantaSoft software (Bio-Rad Laboratories, Hercules, CA, USA). A valid result was considered when more than 10,000 droplets were generated in each reaction.

All experiments included 20ng of a positive control DNA from *F. nucleatum* subsp. *nucleatum Knorr* (ATCC 2558), negative control of DNA from *E. coli* [19,33], and a No Template Control (NTC). Results were expressed as copies/reaction of total DNA added to the reaction. The cases were classified based on the median amount of *Fn* DNA in all *Fn* positive samples as low/negative or high *Fn* score.

### 2.4. Optimization and Analytical Assessment of ddPCR-Based Fn DNA Detection Assay

For the reaction optimization process, different annealing temperatures were evaluated. Then, we optimized amplification conditions by testing a serial dilution curve of *Fn* DNA in *E. coli* DNA by qPCR to determine the assay’s efficiency. Serial dilutions of a positive control DNA from *F. nucleatum* subsp. nucleatum Knorr (ATCC 2558) were prepared, where point 1 contained 10 ng of *Fn* DNA, points 2 to 6 contained serial dilutions of *Fn* DNA in a background of *E. coli* DNA (1:10, 1:100, 1:1000, 1:10,000, 1:100,000, and 1:1,000,000), and point 7 contained only *E. coli* DNA.

### 2.5. Determination of the Limit of Blank (LoB) and Limit of Detection (LoD) of Fn ddPCR

To determine performance and analytical sensitivity of the assay, the limit of blank (LoB) and the limit of detection (LoD) were calculated as previously reported [34]. The mean and standard deviation (SD) of the copy number values obtained from 40 replicates with negative controls (*E. coli* DNA) were used to calculate the LoB: LoB = mean blank + 1.645 * (SD blank). The limit of detection (LoD) was calculated as the lowest *Fn* concentration likely to be reliably distinguished from the background. For this, three independent replicates of ddPCR of the same serial dilution containing *Fn* DNA in a background of *E. coli* DNA described above were performed. The LoD was calculated as LoD = LoB +1.645 * (SD low concentration sample) [34].

### 2.6. Statistical Analysis

Statistical analysis was performed using the software IBM SPSS Statistics 21 for Windows. Categorical variables were compared using Qui-square and Fisher’s exact test. For all analyses, we considered statistical significance when *p* ≤ 0.05 (two-sided). The univariate analysis was performed to assess associations of *Fn* DNA amount as a two-category variable (*Fn*-low/negative and *Fn*-high) with clinicopathological and molecular features. To test for associations between the presence of *Fn* DNA with overall survival and the amount of *Fn* DNA (*Fn*-low/negative and *Fn*-high), Kaplan–Meier curves were constructed, and the log-rank test was used to assess differences in survival between the two categories.

Accuracy measurements were used to evaluate ddPCR and qPCR’s discriminatory power in detecting *Fn* DNA in FFPE samples. The qPCR results were generated in our previous study [19]. Sensitivity, specificity, positive, and negative predictive values were calculated. Receiver operating characteristic (ROC) curves were constructed, and the area under the curve values was obtained [21]. These ROC curve analyses were considered using the *Fn* status obtained by qPCR in fresh-frozen tissue as the gold standard reference [19].

## 3. Results

### 3.1. Study Population

The clinicopathological and molecular features of the 139 CRC patients evaluated in this study were previously reported [19] and are summarized in Table 1. Patient follow-up was recently updated (March 2021) to allow for a survival analysis.

### 3.2. Optimization of ddPCR Assay and Determination of Fn ddPCR LoB and LoD

The annealing temperature optimization in ddPCR showed an optimal reaction temperature of 60 °C (Appendix A). We also evaluated the primer and probe efficiency through serial dilution points. The detection by qPCR showed the amplification of the *Fn* DNA in the first six dilution points, with *Fn* DNA concentrations of 1:10, 1:100, 1:1000, 1:10,000, 1:100,000, and 1:1,000,000 consecutively for each dilution point. No amplification was observed at the sample with only *E. coli* DNA) (Appendix A). Based on the results obtained through the qPCR standard curve, an R^2^ of 0.998 and efficiency of 97.133% were observed (Appendix A).

By measuring 40 negative control replicates, we determined the LoB as 0.5 copies/reaction (Figure 1A). Therefore, all cases with copies/reaction above this value were considered positive for *Fn* DNA. Next, the linearity of the assay and the analytical sensitivity of the reaction was evaluated by serial dilutions. A high linearity between the expected and the measured fraction of *Fn* DNA were observed with an R^2^ = 0.9987 (Figure 1B). The lowest concentration of sample dilution replicates above the LoB were used to determine the LoD (2.10^−3^ ng), leading to a 2.7 copies/reaction value.

### 3.3. Detection of Fn DNA by ddPCR and Comparison with qPCR

Based on the LoD, a sample was considered positive for *Fusobacterium nucleatum* DNA when it had more than 2.7 copies/reaction. Overall, of the 139 CRC FFPE samples evaluated, 38.8% (54/139) were positive for the presence of *Fn* DNA by ddPCR (Appendix A).

Next, the *Fn* levels measured by ddPCR were compared with those previously reported by quantitative PCR (qPCR) in the same FFPE CRC cases [19]. Based on the median number of copies/reaction obtained from the cases exhibiting a presence of *Fn* DNA (median = 69), cases were scored as *Fn* high (above the median) or low/negative (below the median). Therefore, the results showed 20.1% (28/139) of samples as *Fn* high by ddPCR (Figure 2A and Table 2). Similarly, using the same classification of high (above median) or low/negative (below median), *Fn* DNA detected by qPCR were classified and, as reported [19], 2.9% (4/139) were considered *Fn* high (Figure 2B and Table 2). Of note, all four *Fn*-high qPCR cases were also high with the use of ddPRC. The Cohen’s Kappa test showed no correlation (0.210; Table 3) between both methods for *Fn* detection.

### 3.4. Accuracy Measurements for Fn Detection

We further evaluated the accuracy of ddPCR and qPCR for *Fn* DNA detection in FFPE CRC samples, considering the previously reported *Fn* status obtained by qPCR in fresh-frozen tissue as the gold standard [19]. *Fn* DNA were detected by qPCR in 24.4% (34/139) of fresh-frozen tumor samples [19]. ROC curves were generated, and the AUC values were determined to describe the accuracy of ddPCR and qPCR for the *Fn* DNA detection in FFPE CRC samples. The AUC value for *Fn* detection by ddPCR was 0.779 (95% confidence interval (CI) 0.678–0.879) (Figure 3), with a sensitivity of 70.6% and specificity of 80.0%, a positive predictive value of 53.3%, a negative predictive value of 89.4%, and an accuracy rate of 77.7%. Due to the low number of high-*Fn* cases identified when testing by qPCR and the ROC curve, the AUC levels could not be determined (Appendix A).

### 3.5. Association between Fn Status and Patients’ Clinical-Pathological and Molecular Features

The association of *Fn* score detection by ddPCR and qPCR with patients’ clinicopathological and molecular features was also evaluated (Table 2). The high amount of *Fn* DNA detected by ddPCR was associated with proximal tumor location (*p* = 0.04), poorly differentiated histologically tumors (*p* = 0.02), MSI-high status (*p* < 0.001), *BRAF*-mutated tumors (*p* = 0.003), and with the loss of expression of mismatch repair proteins MLH1 (*p* < 0.001), and PMS2 (*p* < 0.001) (Table 2). Concerning the results obtained by qPCR, we found an association only with *BRAF*-mutated tumors (*p* = 0.03) in high *Fn* DNA (Table 2).

Moreover, we interrogated the impact of *Fn* on patient survival. Patients who had a high amount of *Fn* DNA detected by ddPCR had a shorter overall survival when compared to patients with a low/negative amount of *Fn* DNA, yet it did not reach statistical significance (58.3% vs. 76.8% at five years; log-rank *p* = 0.15) (Figure 4A). No significance was observed when analyzing *Fn* DNA by qPCR (60.0% vs. 75.0% at 5 years; log-rank *p* = 0.31; Figure 4B).

## 4. Discussion

The present study reports the optimization and implementation of a ddPCR-based methodology to detect and quantify intra-tumoral *F. nucleatum* in FFPE tissues. We found that 20% of CRC cases showed a high amount of *Fn,* whereas only 2.9% of the cases were *Fn*-high by qPCR. ddPCR, *Fn*-high CRC was associated with known features of *Fn* positive CRC, such as proximal location, poorly differentiated tumors, MSI-High, and *BRAF*-mutated patients.

The use of FFPE tissue to detect microorganisms is challenging, and its accuracy is very dependent on the methodology used [35]. It is well-known that in formalin-fixed tissue, the cross-linking of histone-like proteins to DNA and the fragmentation/degradation of genomic DNA occurs over time, further decreasing the sensitivity of identifying organisms, such as bacteria, using PCR-based approaches [36]; FFPE is frequently the only material available in a clinical setting. The primary assay currently used for *Fn* detection is qPCR, yet we and others reported the limitations of this methodology for FFPE tissues [17,19,21,27,37,38]. Additionally, distinct quantification methods can be used in qPCR, such as using endogenous DNA [21,39], or without the endogenous DNA, through a calibration line [22], complicating data comparison. To circumvent these limitations, other methodologies, such as ddPCR, were developed. ddPCR is an end-point measurement technique that allows for the direct counting of targets without the need for calibration curves, leading to greater accuracy and reproducibility [40]. ddPCR is also less influenced by the presence of PCR inhibitors that bind to DNA, making it unavailable for amplification, or interfering with the DNA polymerase, reducing PCR amplification efficiency; both of these actions underestimate values when using qPCR [41,42]. Additionally, ddPCR is less affected by the PCR amplification efficiency as long as amplification still occurs and the fluorescence signal does not drop below the given threshold [42,43]. The results from the PCR reactions indicate that ddPCR may be advantageous compared to qPCR when dealing with complex samples such as degraded FFPE tissue [42,43].

In the current study, we optimized and implemented a ddPCR-based method for *Fn* DNA detection in FFPE CRC tumor tissue. It is essential to fully characterize the analytical performance of an assay in order to understand its accuracy and limitations [34]. LoB and LoD are necessary for the discrimination between the presence or absence of the target [34]. Accordingly, we showed that a sample needed to have more than 2.7 copies/reaction (LoD) to be considered positive. The ddPCR analysis of FFPE CRC tissue had a similar prevalence of *Fn* positive results (28/139; 20.1%) to fresh-frozen CRC tissue results by qPCR (34/139; 24.4%) [19]. These results agree with previous studies using fresh-frozen CRC tissue, which reported frequencies varying between 8.6% and 87.1% [19,21,27,39,44,45,46].

We observed that a higher presence of intra-tumoral *Fn* by ddPCR in FFPE tissue was associated with several CRC clinical and molecular features, such as proximal tumor location, higher depth of invasion, poorly differentiated tumors, MSI-positive status, *BRAF*-mutated tumors, and the loss of MMR proteins. These results agree with findings from previous studies conducted in other populations, suggesting a role of *Fn* with a subtype of more aggressive CRC for patients with worse prognoses [20,21,44,47,48,49].

We also found that patients with a high amount of *Fn* DNA detected by ddPCR in FFPE had a shorter overall survival than patients with a low/negative amount of *Fn* DNA, the same trend as previous studies [20,21,44,47,48,49]. This association was not observed when we used qPCR in FFPE tissue, most probably due to the low detection of *Fn* high (4/139; 2.9%) by qPCR. In addition, the Cohen’s Kappa test showed no correlation between the results from qPCR and ddPCR (Cohen’s Kappa = 0.210).

There is a critical demand to validate bacterial candidates such as *Fn* for CRC and to investigate their clinical application values by simple and cost-effective quantification methods such as ddPCR [50]. Our study also evaluated the ROC curves and the AUC values to obtain the accuracy of *Fn* detection in FFPE CRC tissue. The AUC value for *Fn* detection by ddPCR was 0.779 (95% confidence interval (CI) 0.678–0.879).

Nowadays, the main disadvantages of using ddPCR are the costs, which are still higher than standard qPCR, the lack of standardized methods, and the limited number of laboratories equipped with instruments [51]. However, some applications, for which ddPCR has a superior performance than qPCR, should be considered for some assays requiring a high precision to measure bacterial load [52].

In conclusion, the present work shows the feasibility of detecting intra-tumoral *F. nucleaum* using the ultrasensitive droplet digital PCR technique. Our findings also highlight the clinical relevance of intra-tumoral *Fn* detection in a routine setting of colorectal cancer.

## Figures and Tables

**Figure 1 diagnostics-12-00114-f001:**
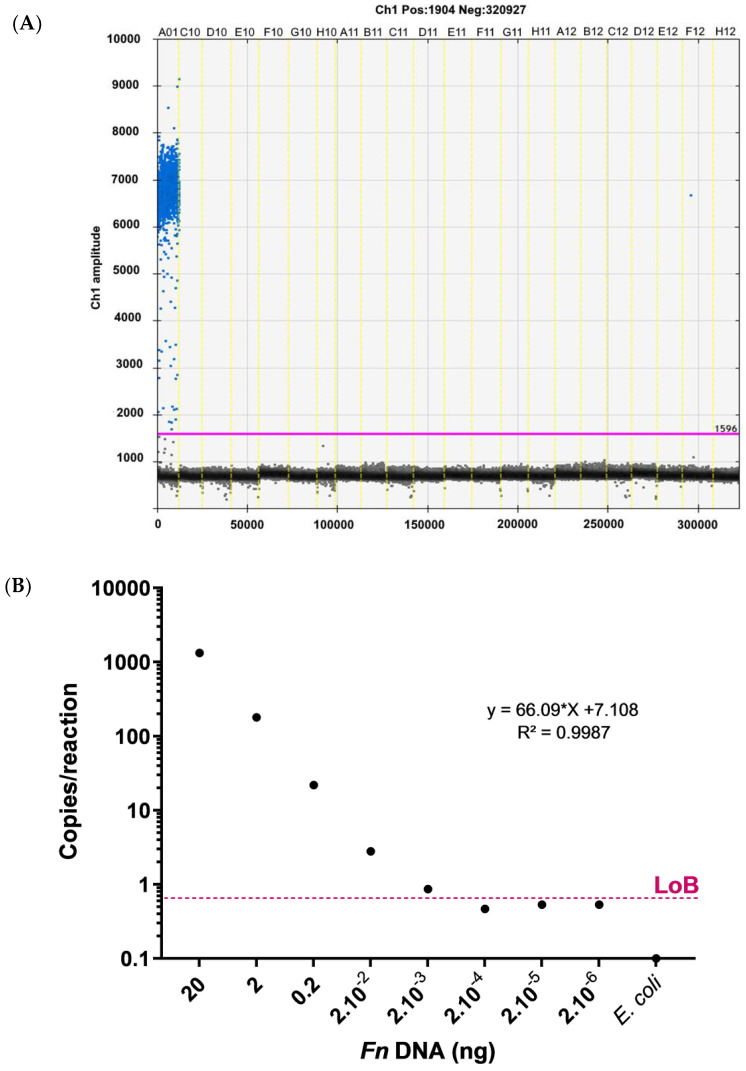
(**A**) The limit of blank (LoB) assay. One-dimensional plots of *Fn* copy numbers by ddPCR. A01: positive control, C10 to F12: *E. coli* DNA, and H12: no template control (NTC). (**B**) Serial dilution of *Fn* DNA in *E. coli*-background DNA. The number of *Fn* DNA copies determined by ddPCR were plotted against the corresponding dilutions.

**Figure 2 diagnostics-12-00114-f002:**
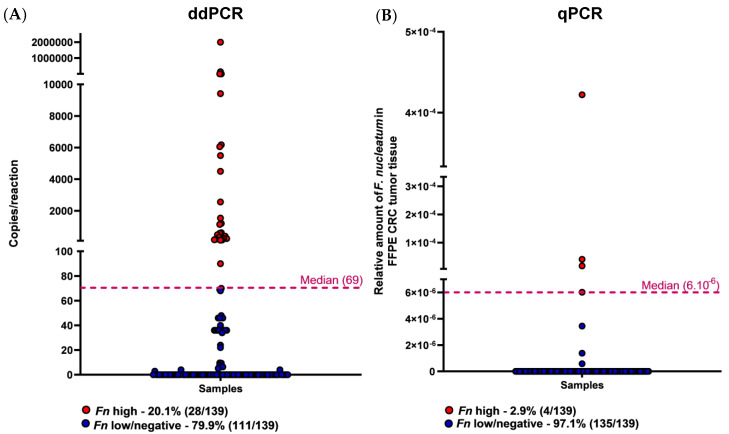
Amount of *F. nucleatum* in 139 CRC FFPE tumor samples according to each methodology. Dotplots represent samples, and the dotted line represents the median cut-point amount around which samples were classified as having a high (above median) or low/negative (below the median) amount of *Fn*. (**A**) The distribution of *Fn* DNA in copies/reaction detected by ddPCR (median = 69 copies/reaction). (**B**) The relative amount distribution of *Fn* DNA was detected by qPCR (median = 6 × 10^−6^).

**Figure 3 diagnostics-12-00114-f003:**
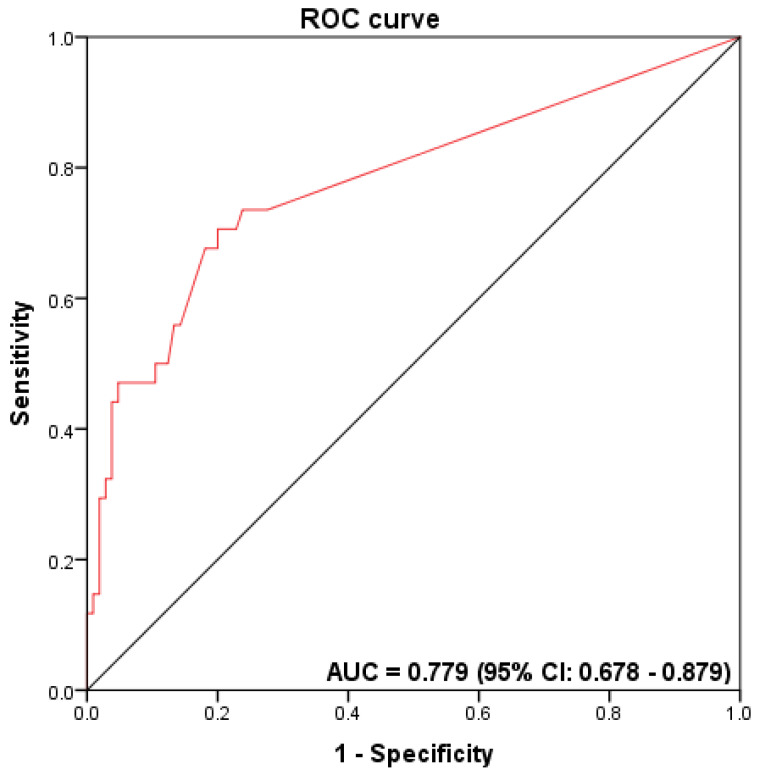
Receiver operator characteristic (ROC) curves to assess the discriminatory accuracy of *Fn* DNA detection by ddPCR.

**Figure 4 diagnostics-12-00114-f004:**
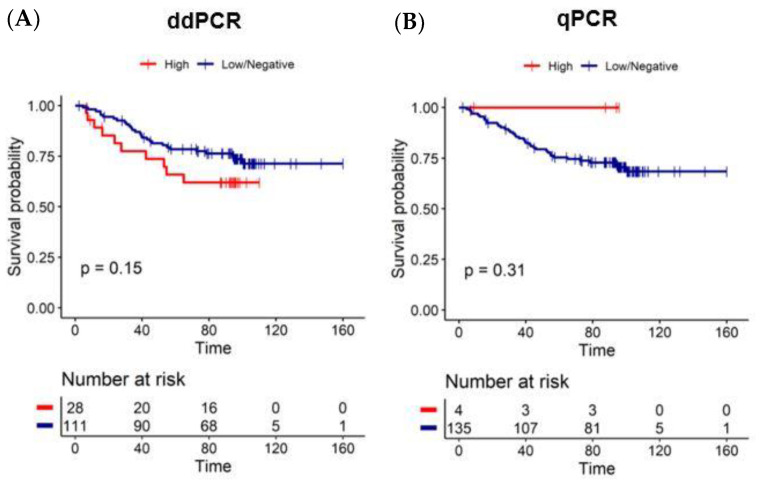
Kaplan–Meier curves for overall survival according to the method detection and the amount of *Fn* DNA and status of the FFPE CRC tissue. (**A**) Five-year overall survival is 58.3% for *Fn* high vs. 76.8% for *Fn* low/negative, detected by ddPCR (log-rank *p* = 0.15). (**B**) Five-year overall survival was 60.0% for *Fn* low/negative vs. 75.0% for *Fn* high detected by qPCR (log-rank *p* = 0.31).

**Table 1 diagnostics-12-00114-t001:** Clinicopathological and molecular features of 139 CRC patients.

Features		Number of Cases (*n*)	(%)
**Mean age 60.65 ± 13.49**			
**Gender**	**Female**	65	46.5
	**Male**	74	53.2
**Clinical stage (at diagnosis)**	**0/I**	36	26.1
	**II/III**	98	69.6
	**IV**	6	4.3
**Tumor location**	**Proximal colon**	38	27.3
	**Distal colon**	72	51.8
	**Rectum**	29	20.9
**Tumor (T)**	**Tis/T1/T2**	44	31.7
	**T3/T4 (a b)**	95	68.3
**Tumor differentiation**	**Well to moderate**	126	92.0
	**Poor**	11	8.0
**MSI status ^#^**	**MSS/MSI-low**	118	84.9
	**MSI-high**	21	15.1
** *BRAF* ** **mutation ^#^**	**Mutant**	11	8.0
	**Wild-type**	127	92.0
**MLH1 protein expression ^#^**	**Positive**	105	86.1
	**Negative**	17	13.9
**MSH2 protein expression ^#^**	**Positive**	119	97.9
	**Negative**	3	2.1
**MSH6 protein expression ^#^**	**Positive**	121	99.3
	**Negative**	1	0.7
**PMS2 protein expression ^#^**	**Positive**	107	87.7
	**Negative**	15	12.3
**Status**	**Alive without cancer**	78	56.1
	**Alive with cancer**	7	5.0
	**Death from cancer**	38	27.3
	**Death from other causes**	15	10.8

^#^ Previously reported by de Carvalho et al. [19]; MSI, microsatellite instability.

**Table 2 diagnostics-12-00114-t002:** Clinicopathological and molecular features according to the amount of *Fusobacterium nucleatum* (*Fn*) DNA in FFPE CRC tissue detected by ddPCR and qPCR.

Variable		All Cases (%)	ddPCR			qPCR		
			**Neg/Low** ***n* = 111 (79.9)**	**High** ***n* = 28 (20.1)**	***p*-Value ***	**Neg/Low** ***n* = 135 (97.1)**	**High** ***n* = 4 (2.9)**	***p*-Value ***
**Age**	**Mean 60.65 ± 13.49**							
**Gender**	**Female**	65 (46.8)	49 (75.4)	16 (24.6)	0.2	65 (48.1)	0 (0.0)	0.1
	**Male**	74 (53.2)	62 (83.8)	12 (16.2)		70 (51.9)	4 (100.0)	
**Tumor location**	**Proximal colon**	38 (27.3)	26 (23.4)	12 (42.9)	**0.04**	36 (26.7)	2 (50.0)	0.6
	**Distal colon**	72 (58.1)	58 (52.3)	14 (50.0)		70 (51.9)	2 (50.0)	
	**Rectum**	29 (20.9)	27 (24.3)	2 (7.1)		29 (21.5)	0 (0.0)	
**Tumor (T)**	**Tis/T1/T2**	44 (31.7)	36 (32.4)	8 (28.6)	0.6	42 (31.1)	2 (50.0)	0.5
	**T3/T4 (a b)**	95 (68.3)	75 (67.6)	20 (71.4)		90 (68.9)	2 (50.0)	
**Clinical stage**	**0/I**	37 (26.6)	30 (27.0)	7 (25.0)	1.0	35 (25.9)	2 (50.0)	0.3
	**II/III**	96 (69.1)	76 (68.5)	20 (71.4)		94 (69.6)	2 (50.0)	
	**IV**	6 ( 4.3)	5 (4.5)	1 (3.6)		6 (4.4)	0 (0.00)	
**Tumor differentiation**	**Well to moderate**	126 (92.0)	104 (94.5)	22 (81.5)	**0.02**	124 (92.5)	2 (66.7)	0.2
	**Poor**	11 (8.0)	6 (5.5)	5 (18.5)		10 (7.5)	1 (33.3)	
**MSI status**	**MSS/MSI-Low**	118 (84.9)	101 (91.0)	17 (60.7)	**<0.001**	116 (85.9)	2 (50.0)	0.1
	**MSI-High**	21 (15.1)	10 (9.0)	11 (39.3)		19 (14.1)	2 (50.0)	
** *BRAF* ** **mutation**	**Mutant**	11 (8.0)	5 (4.5)	6 (21.4)	**0.003**	9 (6.7)	2 (50.0)	**0.03**
	**WT**	127 (92.0)	105 (95.5)	22 (78.6)		125 (93.3)	2 (50.0)	
**MLH1 protein expression**	**Positive**	105 (86.1)	90 (92.8)	15 (60.0)	**<0.001**	103 (89.6)	2 (50.0)	0.09
	**Negative**	17 (13.9)	7 (7.2)	10 (40.0)		12 (10.4)	2 (50.0)	
**MSH2 protein expression**	**Positive**	119 (97.5)	95 (97.9)	24 (96.4)	0.5	115 (97.5)	4 (100.0)	1.0
	**Negative**	3 (2.5)	2 (2.1)	1 (4.0)		3 (2.5)	0 (0.0)	
**MSH6 protein expression**	**Positive**	121 (99.2)	96 (99.0)	25 (100.0)	1.0	117 (99.2)	4 (100.0)	1.0
	**Negative**	1 (0.8)	1 (1.0)	0 (0.0)		1 (0.8)	0 (0.0)	
**PMS2 protein expression**	**Positive**	107 (87.7)	91 (93.8)	16 (64.0)	**<0.001**	105 (87.3)	2 (50.0)	0.07
	**Negative**	15 (12.3)	6 (6.2)	9 (36.0)		15 (12.7)	2 (50.0)	

The percentage indicates the proportion of cases with a specific clinicopathological or molecular variable according to the amount of *Fn* DNA in the FFPE CRC tissue.* For the association between the amount (Negative/low vs. high) of *Fn* DNA in CRC FFPE tissue, Fisher’s exact test was performed.

**Table 3 diagnostics-12-00114-t003:** Concordance of *Fusobacterium nucleatum* (*Fn*) detection in FFPE CRC cases by ddPCR and qPCR methods.

	Concordance Rate % (*n*)	Cohen’s Kappa
FFPE ddPCR × FFPE qPCR	82.8% (115/139)	0.210

FFPE: formalin-fixed, paraffin-embedded.

## Data Availability

The datasets generated for this study are available on request from the corresponding authors.

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
