# Peer review of "Accuracy and Clinical Relevance of Intra-Tumoral Fusobacterium nucleatum Detection in Formalin-Fixed Paraffin-Embedded (FFPE) Tissue by Droplet Digital PCR (ddPCR) in Colorectal Cancer"

_diagnostics, 2022, doi:10.3390/diagnostics12010114_

Round 1

Reviewer 1 Report

This article describes the establishment of ddPCR method to detect Fusobacterium nucleatum (Fn) DNA in formalin-fixed paraffin-embedded (FFPE) CRC. The authors evaluated Fn DNA detection's accuracy and clinical relevance by ddPCR. Measures of accuracy for Fn status compared to prior results generated by qPCR and association with clinicopathological and molecular patients’ features were also evaluated. To improve the integrity, I have several questions:

  1. As the authors described, the Fn DNA in 139 CRC FFPE cases was measured by qPCR previously. There was a low agreement between Fn status determined by ddPCR and qPCR. The qPCR method was poorly described in the article, which made me wonder its performance and fitness. It would be more convincing to characterize qPCR method in more detail.
  2. The use of FFPE tissue to detect microorganism is challenging. DNA extraction and purification efficiency influence the quantitative results greatly. The results generated by qPCR were gained previously, I wonder whether the low agreement of ddPCR and qPCR results was due to the different DNA extraction procedure partially.
  3. The authors talked that histone-like proteins and DNA were cross-linked in formalin-fixed tissue, which was thought to decreasing sensitivity of PCR-based detection method. As we know, the cross-linked DNA and protein can be reversed at 65℃ incubation. I wonder whether reversing cross-link can improve the performance of ddPCR and qPCR, or their agreement.
  4. When discussing the impact of Fn DNA content on patient survival. Have you consider about the difference of medical treatment among the patients?

Reviewer 2 Report

The paper by Datorre et al. analyzes the presence of Fusobacterium nucleatum in colorectal cancer cases focusing on the detection in formalin-fixed paraffin-embedded (FFPE) tissue by droplet digital PCR (ddPCR).

The correlations between the 28 cases with high FN and molecular and clinicopathological features, as MSI status, BRAF mutation, MLH1 and PMS2 protein expression, are very interesting (Table 2).

The authors also compare the FN DNA detection by ddPCR in this study to FN DNA detection by qPCR in the same series of patients, both in frozen tissue than in FFPE, analyzed in a previous paper by de Carvalho et at. Front Oncol 2019.

  1. Along all the manuscript the authors compare the two methods, not always clearly.

F. nucleatum, initially considered a passenger bacterium in human intestinal tract, is now considered to be a potential initiator of CRC susceptibility. F. nucleatum invades human epithelial cells, activates β-catenin signaling, induces oncogenic gene expression and promotes growth of CRC cells through the FadA adhesion virulence factor. Increasing evidence indicated that the levels of F. nucleatum are significantly elevated in tumor tissues of CRC patients relative to normal controls and that F. nucleatum is considered a potential risk factor for CRC progression. Previous papers have demonstrated that a higher abundance of F. nucleatum in CRC is associated with staging and a shorter survival time.

The limitations of the ddPCR have been underlined by the authors.

As regards applications, DNA extraction from paraffin tissue has always been undoubtedly more difficult than fresh frozen tissue. This has led to developing a third generation of not conventional polymerase chain reaction, ddPCR, to directly quantify and clonally amplify DNA.

This method is preferably used in low-abundance nucleic acid detection, as in the case of FFPE or to identify rare alleles in a heterogeneous tumor. If you compare it with qPCR, the latter has the limitation of the instability in low concentration template amplification.

In the case of frozen tissue, on the other hand, qPCR is optimal in detecting bacterial DNA, such as Fusobacterium. Furthermore, the detection can be carried out without comparing it to an endogenous DNA, as some authors do, such as Castellarin et al. Genome Res 2012, but by constructing a calibration line, as other authors do such as for example Pignatelli et al. Cancers (Basel). 2021 DOI: 10.3390/cancers13051032

2. Please discute this point in the text of the manuscript.

Results

3. In lines 73/74 was stated that ddPCR exibit much great sensitivity than qPCR, but in lines 194/195 is stated that there is no correlation between FN detected with the two methods. Does this mean that the 4 cases above the median detected with qPCR did not fall in the 28 cases above the median in ddPCR? If is so how can be explained?

4. In lines 122/126 the procedure for assay optimization condition was described, where 20 ng of FN DNA was diluted in E.coli DNA. Was 20 ng of E.Coli DNA the solution for making serial dilution? Was a similar curve done using simply water? If yes were the two curves similar?

5. in line 193/195 was stated that FN DNA detected by qPCR was reclassified on the basis of results obtained by ddPCR. Could you better explain how the qPCR reclassification was done. Maybe does it mean that the High FN in the paper by de Carvalho et al. 2019 were 5 and in this study they are 4?

6. Paragraph 3.4 Accuracy measurements for Fn detection

in the first line, “compared” is to be removed because the comparison between the two methods was not possible (“when testing by qPCR, ROC curve and AUC levels could not be determined”.

Figures

7. A and B are lacking in Figure 1 and Figure 2

Discussion

The authors wrote:

“We also found that patients with a high amount of Fn DNA detected by ddPCR had shorter overall survival than patients with a low/negative amount of Fn DNA, the same trend as previous studies[20, 36, 38, 44, 48-50]. However, in contrast to results by qPCR, it was demonstrated that patients with a low/negative amount of Fn DNA had shorter overall survival, and this goes on the opposite way to other studies [19-21, 44-47], one possible explanation is due to the low detection of Fn high (4/139; 2.9%) by qPCR. In addition, the Cohen’s Kappa test demonstrated an absence of correlation between results from qPCR and ddPCR (Cohen’s Kappa = 0.210).”

8. Better explain the sentence:

“However, in contrast to results by qPCR, it was demonstrated that patients with a low/negative amount of Fn DNA had shorter overall survival, and this goes on the opposite way to other studies”

but in the last paragraph of Results the authors wrote: “No significance was observed when analyzing Fn DNA by qPCR (60.0% vs 75.0% at 5 years; log-rank p = 0.31”

In FFPE or frozen tissue?? In this study or the previous (2019)?

Maybe it does mean that ddPCR is more accurate than qPCR as this study better stratifies patients with High and Low FN DNA but it is also true that FN abundance measured by qPCR overall is associated with staging (Pignatelli et al. Cancers 2021).

References

9. In the legend of Table 1 ref 9 is wrong

In ref 12 the name of the journal is lacking

There are no references on the use of qPCR in CRC (such as for example Pignatelli et al. Cancers 2021)
